health and disease and epidemiology/ computational biology/ecology

livestock movements, cattle, disease, contact chains, *Mycobacterium bovis*

**Author for correspondence:**
Robbie A. McDonald
e-mail: r.mcdonald@exeter.ac.uk

# Effects of trading networks on the risk of bovine tuberculosis incidents on cattle farms in Great Britain

Helen R. Fielding[1,3], Trevelyan J. McKinley[2], Richard J. Delahay[4], Matthew J. Silk[1] and Robbie A. McDonald[1]

[1]Environment and Sustainability Institute, and [2]College of Engineering, Mathematics and Physical Sciences, University of Exeter, Penryn Campus, Penryn TR10 9FE, UK
[3]The Royal (Dick) School of Veterinary Studies and The Roslin Institute, University of Edinburgh, Easter Bush Campus, Midlothian EH25 9RG, UK
[4]Animal and Plant Health Agency, Woodchester Park, Nympsfield, Stonehouse GL10 3UJ, UK

HRF, 0000-0001-7041-1380; TJM, 0000-0002-9485-3236;
RJD, 0000-0001-5863-0820; MJS, 0000-0002-8318-5383;
RAM, 0000-0002-6922-3195

Trading animals between farms and via markets can provide a conduit for spread of infections. By studying trading networks, we might better understand the dynamics of livestock diseases. We constructed ingoing contact chains of cattle farms in Great Britain that were linked by trading, to elucidate potential pathways for the transmission of infection and to evaluate their effect on the risk of a farm experiencing a bovine tuberculosis (bTB) incident. Our findings are consistent with variation in bTB risk associated with region, herd size, disease risk area and history of previous bTB incidents on the root farm and nearby farms. However, we also identified effects of both direct and indirect trading patterns, such that connections to more farms in the England High-Risk Area up to three movements away from the root farm increased the odds of a bTB incident, while connections with more farms in the England Low-Risk Area up to eight movements away decreased the odds. Relative to other risk factors for bTB, trading behaviours are arguably more amenable to change, and consideration of risks associated with indirect trading, as well direct trading, might therefore offer an additional approach to bTB control in Great Britain.

## 1. Introduction

Understanding pathways for the transmission of infections is fundamental to devising efficient control strategies for livestock diseases. These pathways exist at many scales, describing transmission among individuals [1], farms [2] and countries [3].

Trading connections among farms can generate vast networks of animal movements [4], which have been implicated in disease transmission [5,6]. Network epidemiology uses these networks as a framework on which to model the spread of infection [7] and measures describing the centrality of a farm in a trading network and the nature of its contact chains (describing temporal connectedness) can be useful in assessing an individual farm's risk of acquiring infection [8].

*Mycobacterium bovis* causes bovine tuberculosis (bTB) in cattle but can infect a wide range of mammalian species [9]. Despite intense study and significant resources invested in control measures, the infection remains endemic in large parts of Great Britain [10]. Among other factors, wildlife reservoirs of infection, most notably in badgers *Meles meles*, and undetected infections and movements of cattle have been shown to contribute to disease persistence and spread. A dynamic transmission model using British cattle movement and bTB testing data from 1996 to 2011 predicted that the movements of cattle alone accounted for 13% of bTB incidents and played a further role in multifactorial bTB incidents [11]. Modelling by Green *et al.* [12] suggested that in 2004, 16% of herd infections occurred as a result of cattle movements and that local effects were more important. The effects of cattle movements on patterns of bTB incidence in Great Britain have varied across regions [13,14] and over time as new policies have been implemented [15]. However, in spite of the introduction of pre-movement and, in some limited circumstances, post-movement testing for animals leaving high-incidence areas [15–17], studies using data from 2006 to 2013 still suggested that there was an increased risk of bTB incidents for herds that had purchased cattle from high-risk regions of England and Wales [17–19].

When compared to other risk factors for bTB incidents, such as location and herd type, cattle movements represent an activity that might be more amenable to management, for example, by the use of risk-based trading, regulation or legislation. While the role of direct cattle movements in the transmission of bTB in Great Britain has been well-documented, the risks associated with indirect movements arising from trade have not yet been quantified or explored in detail. Second-order contacts (i.e. contacts of contacts) in farm networks have been shown to improve the estimates of simulated epidemic size in models of disease in the British cattle herd [20] and, in Italy, influence the roles of individual farms in disease transmission [21]. In order to interrogate the farm trading network beyond first-order, direct contacts, data on the sequential purchases of cattle can be used to construct temporally explicit contact chains among farms. The ingoing contact chain (ICC) represents the source farms that may contribute infection to the root farm, while the outgoing contact chain represents those farms to which the root farm might transmit infection. The magnitude of ingoing contact chains has been associated with the risk of bTB in French cattle [8] and was used to inform risk-based targeting of farms for surveillance of bovine coronavirus and bovine respiratory syncytial virus in Sweden [22].

We have previously constructed contact chains for all cattle herds in Great Britain and found that a large proportion of herds have remarkably extensive chains, extending to tens of thousands of other farms within 12-month periods [4]. In the present study, we hypothesize that farms with larger numbers of farms in their ICCs are at greater risk of exposure and acquisition of infection. We predict that infection risks might therefore be greater for farms with ICCs that include more risky trading partners, such as those that have experienced a recent bTB incident or that are located in regions with high incidence rates. We also predict that the closer a farm is to the root farm, both in terms of geographical proximity and trading proximity, the more impact it will have on disease risk on the root farm. It has been established that risk factors are likely to vary across different geographical regions [14,23]. Therefore, we performed multiple analyses using subsets of data from specific regions. Overall, our study aimed to assess the importance of new parameters from contact chains based on trading networks, alongside established risk factors, on the risk of bTB incidents on cattle farms in Great Britain.

## 2. Material and methods

### 2.1. Study farms and bovine tuberculosis data

We obtained bTB surveillance and farm information data collated in the Cattle Tracing System (CTS) by the Animal and Plant Health Agency (APHA). Study farms were active premises (i.e. registered cattle holdings with a birth, death or movement) between 1 January 2012 and 31 December 2014 with full bTB testing and location data, resulting in a final dataset comprising 71 096 cattle farms. Annual herd size was derived from the mean daily number of cattle on the premises. We took the mean of these annual herd sizes across the three years of the study. Herd type was defined using CTS data, first by the predominant breed type (dairy, beef or dual purpose) and then sex of animals on the farm (as outlined in [4]). Farms were allocated to bTB risk areas for England based on the designated risk area of their county parish defined

by APHA in 2015 [24] (electronic supplementary material, methods and figure S1). A farm was defined as having had a bTB incident if its officially tuberculosis free (OTF) status was withdrawn (OTF-W) or suspended (OTF-S) on one or more days during the time period in question, i.e. for assessing previous bTB history this period was 2010–2014, and for our response variable, it was 2015–2016. OTF status is suspended (OTF-S) if at least one animal tests positive (or returns two inconclusive test results) for *M. bovis*, using either of the statutory live tests: the single intradermal cervical comparative tuberculin (SICCT) test or the gamma interferon test, and is withdrawn (OTF-W) upon finding *post-mortem* pathology characteristic of *M. bovis* or isolation of the bacteria by culture. Both types of incident were included in the analysis in order to maximize the power of tests of the association between infection in the contact chain and subsequent incidents. Two additional analyses were performed with each type of incident as the response variable (OTF-S and OTF-W), but final outcomes did not change substantially and so the simpler, combined analysis is presented here.

## 2.2. Networks

A single-directed network was calculated for each of the three study years, representing farms as nodes and with farm-to-farm movements of at least one animal represented by edges, weighted by the number of animals traded between those farms in the study year (electronic supplementary material, methods). Betweenness centrality was calculated for each root farm in each network, providing a centrality measure of the importance of the farm in connecting together different parts of the network [25]. ICCs are comprised of farms linked by movements of animals *onto* farms in a chronological sequence, such than at least one animal from a movement could have potentially seeded an infection that could, in principle, be passed on in a subsequent movement of animals from that farm. Any farm that may have contributed to infection on the root farm (via movements) within the study period is included in the chain as a 'source farm'. The 'level' for each source farm corresponds to the minimum number of movements away from the root farm at which they appear in the ingoing contact chain. To avoid any seasonal effect of starting the chain in a specific month, we calculated 24 ICCs at one month intervals, each spanning 12 months of movements. We then grouped the source farms from each of the 24 ICCs. Where farms appeared more than once in the chain in any one year, we included them at the closest (minimum) level to the root farm and removed other instances. Each chain was curtailed at eight levels away from the root farm due to computational limitations (electronic supplementary material, methods).

For each root farm, we quantified how many source farms from each region were at each level of the contact chain and how many source farms at each level experienced a bTB incident from 2010 to 2014. We used cumulative counts across the different levels, e.g. the number of source farms at level one, the sum of the number of source farms at levels one and two, etc., up to the eighth level. We recorded the mean distance between each root farm and all of their source farms at each level. To incorporate bTB risks from local farms, we calculated the proportion of farms that had experienced a bTB incident between 2010–2014, within a radius of 0–2 km, 0–4 km, 0–6 km, 0–8 km and 0–10 km of the root farm. Location was defined by the (x, y) coordinates associated with a particular holding identity (County Parish Holding number) in the APHA data. There are limitations with this proxy for farm location because, in some cases, the point location of the CPH does not lie within farmland where cattle are mostly located. However, there is no known systematic spatial bias to these locations, sample size is very large and there are no validated alternatives at this scale. Thus, this is the best proxy to use, and it has been used in previous studies [13,18]. We selected baseline variables for a regression model based on previously identified risk factors [19,23] listed in table 1.

We specified a multivariable regression model using a baseline set of variables (table 1) with the binary response of whether the root farm experienced a bTB incident in 2015 or 2016, i.e. immediately after the period over which the ICCs had been calculated. Various risk factors could be specified at different lags (e.g. distances or levels of the network). However, the inclusion of multiple variables corresponding to a single-risk factor specified at different lags results in high multicollinearity and thus model instability. Instead, we chose a single representative variable within each set of lagged variables by adding each lagged variable in turn to this baseline model and choosing the one with the lowest value of the Akaike information criterion (AIC) [26]. This avoided the inclusion of a large number of highly correlated variables and problems with inference caused by singularity errors in the design matrix, while preventing excessive model selection and therefore reducing bias in the estimated coefficients of the model [27]. As an example, to select the most suitable lag for the risk of bTB from the local area, we ran five multivariable regression models including each distance with the baseline risk factors described above (electronic supplementary material, figure S2). The model with the lowest

**Table 1.** Effect sizes of explanatory variables on the odds of a bovine tuberculosis incident on the root farm in 2015–2016. Odds ratios with 95% confidence intervals are from our multivariable logistic regression analysis using the full Great Britain dataset. Odds ratios of continuous variables are standardized as the odds associated with the difference between the 10th and 90th percentiles of the raw data.

| region | parameter | | 10th percentile (raw data) | 90th percentile (raw data) | odds ratio | 2.5% confidence limit | 97.5% confidence limit |
|---|---|---|---|---|---|---|---|
| Great Britain (n = 71096) | root farm risk area/country | Scotland | — | — | Baseline | | |
| | | Wales | — | — | 6.67 | 5.19 | 8.68 |
| | | England Low-Risk Area | — | — | 2.88 | 2.23 | 3.77 |
| | | England Edge Area | — | — | 10.58 | 8.24 | 13.76 |
| | | England High-Risk Area | — | — | 8.94 | 6.90 | 11.75 |
| | root farm herd type | Mixed | — | — | Baseline | | |
| | | Dairy | — | — | 1.33 | 1.19 | 1.49 |
| | | Fattening | — | — | 0.90 | 0.80 | 1.02 |
| | | Suckler | — | — | 1.07 | 0.97 | 1.19 |
| | root farm bTB incident 2010–2014 (binary) | | — | — | 2.79 | 2.62 | 2.98 |
| | cattle purchased by root farm | | — | — | 0.98 | 0.90 | 1.07 |
| | mean number of farms in ICC | 1st quartile (0–1) | — | — | Baseline | | |
| | | 2nd quartile (2–662) | — | — | 1.04 | 0.94 | 1.15 |
| | | 3rd quartile (663–6280) | — | — | 1.12 | 1.01 | 1.24 |
| | | 4th quartile (6281–39676) | — | — | 1.17 | 0.99 | 1.38 |
| | cattle purchased direct from England High-Risk Area | | — | — | 1.23 | 1.12 | 1.34 |
| | root farm herd size | | 4 | 280 | 19.41 | 17.09 | 22.06 |
| | mean number of purchased cattle | | 0 | 198 | 1.01 | 1.00 | 1.03 |
| | root farm betweenness | | 0 | 304 239 | 1.00 | 0.99 | 1.00 |
| | proportion of farms within 8 km with bTB 2010–2014 | | 0 | 0.54 | 7.36 | 6.51 | 8.33 |
| | no. farms in England High-Risk Area at levels 1–3 | | 0 | 627 | 1.11 | 1.07 | 1.15 |
| | no. farms in England Low-Risk Area at levels 1–8 | | 0 | 10 144 | 0.69 | 0.58 | 0.83 |

AIC included the proportion of farms with bTB during 2010–2014 within a radius of 8 km from the farm, and so this term was included in the final model. This process was repeated to select representative variables from two sets of contact chain variables; those we expected to be associated with increased or decreased risk of a bTB incident on the root farm (electronic supplementary material, figure S3). The final model included all baseline explanatory variables, the proportion of farms in a radius of 0–8 km with a recent bTB incident and the selected representative source farm variables: the number of farms located in the England High-Risk Area at levels 1–3, and the number of farms located in the England Low-Risk Area at levels 1–8 of the ICC (table 1). To capture potential differences in aetiology between different regions, we grouped our root farms by their region (electronic supplementary material, figure S1) and performed the same multivariable analysis, only removing region as an explanatory variable.

To aid comparison of odds ratios (ORs) between variables specified on different scales, we re-scaled the coefficients for continuous variables such that each estimate corresponds to the OR for a variable changing from the 10th percentile to the 90th percentile of the observed data (table 1; electronic supplementary material, table S2). To make it clear that these do not relate to a per-unit increase odds ratio, we henceforth refer to them as *standardized ORs* (note that these are not standardized in the usual statistical sense (using standard deviations), which would not be sensible here due to high asymmetry in the distributions of some of the explanatory variables). Model performance was assessed (electronic supplementary material, figure S4) and we tested for overfitting of the final model using bootstrapping [27]. Given the size of the dataset, the model showed negligible evidence of any overfitting, so we chose to present the unadjusted results here. All analysis was performed using R [28]; network measures were calculated in 'igraph' [29], contact chains were calculated using 'EpiContactTrace' [30], and regression was performed using 'lme4' [31].

## 3. Results

The majority of source farms were located within the same bTB risk area or region as the root farm. However, the mean distance between root and source farms increased with the number of levels in the ICC (electronic supplementary material, figure S5). An average root farm in the England Edge Area, England Low-Risk Area or in Wales had about 20–25% of its source farms located in the England High-Risk Area (figure 1). Also, 21% of root farms in Scotland had over 20% of source farms located in the England High-Risk Area (figure 1). Apart from Scotland, all regions had some farms that had almost all of their source farms located in other regions (figure 1). The maximum number of farms at each level increased up to level 3 and declined thereafter, likely demonstrating a saturation effect where farms connecting at 'higher' levels in the chain had already been counted at 'lower' levels and, therefore, did not increase the total number of farms in the chain. By contrast, the median values for the number of farms increased at each level up to level 8, likely demonstrating an overall amplification effect where the presence of more farms at each level allowed more farms to connect at the next (figure 2).

In 2015–2016, 12.9% of root farms experienced a bTB incident, of which 78.5% were classified as OTF-W (electronic supplementary material, table S1). Proportions of bTB incidents classified as OTF-W or OTF-S in the England High-Risk Area and Wales mirrored those in Great Britain as a whole, whereas in the England Edge Area the relative proportions of these classifications were more equal and in the England Low-Risk Area and Scotland, there were more bTB incidents classed as OTF-S than OTF-W (electronic supplementary material, table S1). Characteristics of the root farm and local bTB history were major risk factors associated with the likelihood of a bTB incident. Compared to farms in Scotland and the England Low-Risk Area, root farms located in the England Edge Area or England High-Risk Area, or in Wales, had greater odds of experiencing a bTB incident in 2015–2016 (figure 3). Root farm herd size was a strong predictor of bTB incidents with the odds of larger herds (the 90th percentile) having a bTB incident in 2015–2016 being 19.41 (95% CI = 17.09–22.06) times higher than for smaller herds (the 10th percentile— table 1). Root farms that had themselves experienced a bTB incident in the period 2010–2014 had 2.79 (2.62–2.98) times higher odds of experiencing another bTB incident in 2015–2016 (table 1). Suckler, fattening and mixed herds had similar odds of a bTB incident, yet the odds of a dairy farm having a bTB incident in 2015–2016 were on average 1.33 (1.19–1.49) times higher than for a mixed farm (table 1). Root farms with a higher proportion of farms within 8 km that had experienced a bTB incident (2010–2014) had 7.36 (6.51–8.33) times higher odds of experiencing an incident themselves (table 1).

In selecting our variables, model performance was better (i.e. a lower AIC) when we included the number of source farms in the England High-Risk Area, than those in the England Edge Area, Wales or the number of source farms with a bTB incident in the last 5 years (electronic supplementary

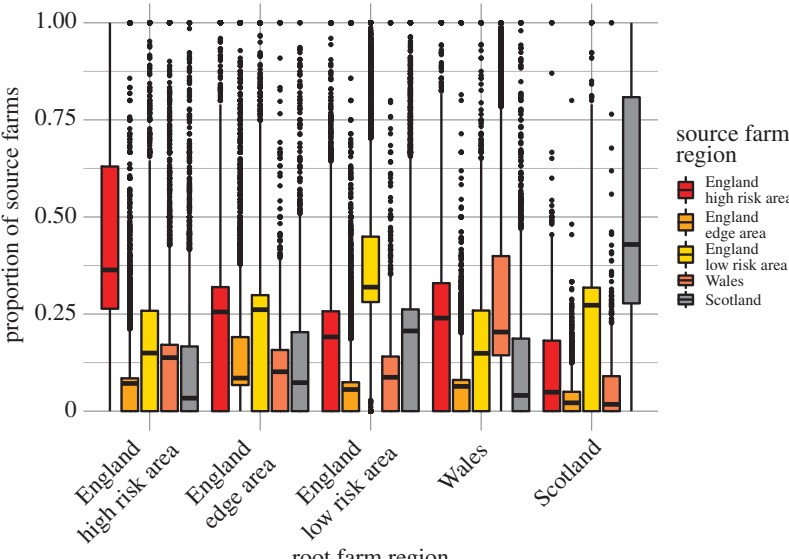

**Figure 1.** Proportional contribution of each bovine tuberculosis risk region to the numbers of source farms comprising the ingoing contact chain of root farms in each of the disease risk regions in Great Britain. The majority of source farms are located within the root farm's region; however, in the England Edge Area, England Low-Risk Area and Wales over 25% of source farms are from the England High-Risk Area. Boxplots show the median, 25th and 75th percentiles, and the upper and lower whiskers extend to the largest or smallest value no further than 1.5 times the interquartile range, data beyond this range are plotted as outlying points.

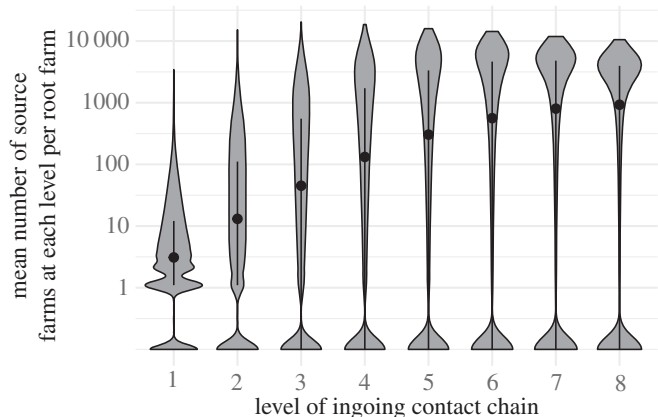

**Figure 2.** Structure of the ingoing contact chains (ICCs) of root farms in Great Britain constructed from cattle trades showing the mean number of source farms at increasing number of movements (levels) away from the root farm (grey violin plots). Black dots indicate the median of the mean number of source farms at each level of the ICC and black lines show the interquartile range. As chain levels increase, more farms are incorporated in the chain up until level 3 where the maximum number of farms starts to decline, likely as a result of a saturation effect. Median values continue to increase throughout all levels, likely showing an amplification effect as more farms are connected at each level. Farms with no source farms are removed from the plot. Data are $n + 0.1$, for depiction on log axis.

material, figure S3). A root farm with more farms in the England High-Risk Area at levels 1–3 of their ICC had 1.11 (1.07–1.15) times higher odds of having a bTB incident (table 1). We found that having more farms in lower incidence areas at any level of the ICC was associated with a decreased risk of bTB incident on the root farm, specifically root farms had lower odds of a bTB incident in 2015–2016 (OR = 0.69, 0.58–0.83; table 1) if they had more source farms located in the England Low-Risk Area at levels 1–8 of their ICC. The odds of having a bTB incident were 1.23 (1.12–1.34) times higher when purchasing cattle direct from the England High-Risk Area compared to not purchasing cattle from the England High-Risk Area (figure 3).

Our regional analyses showed broadly similar effect sizes to those obtained using the entire Great Britain dataset; however, there were key differences in the effect sizes for bTB history on root farms and neighbouring farms (table 1; electronic supplementary material, table S2, and figure 3 for analysis of all factors). In terms of network factors, having more farms in the ICC in the England Low-Risk

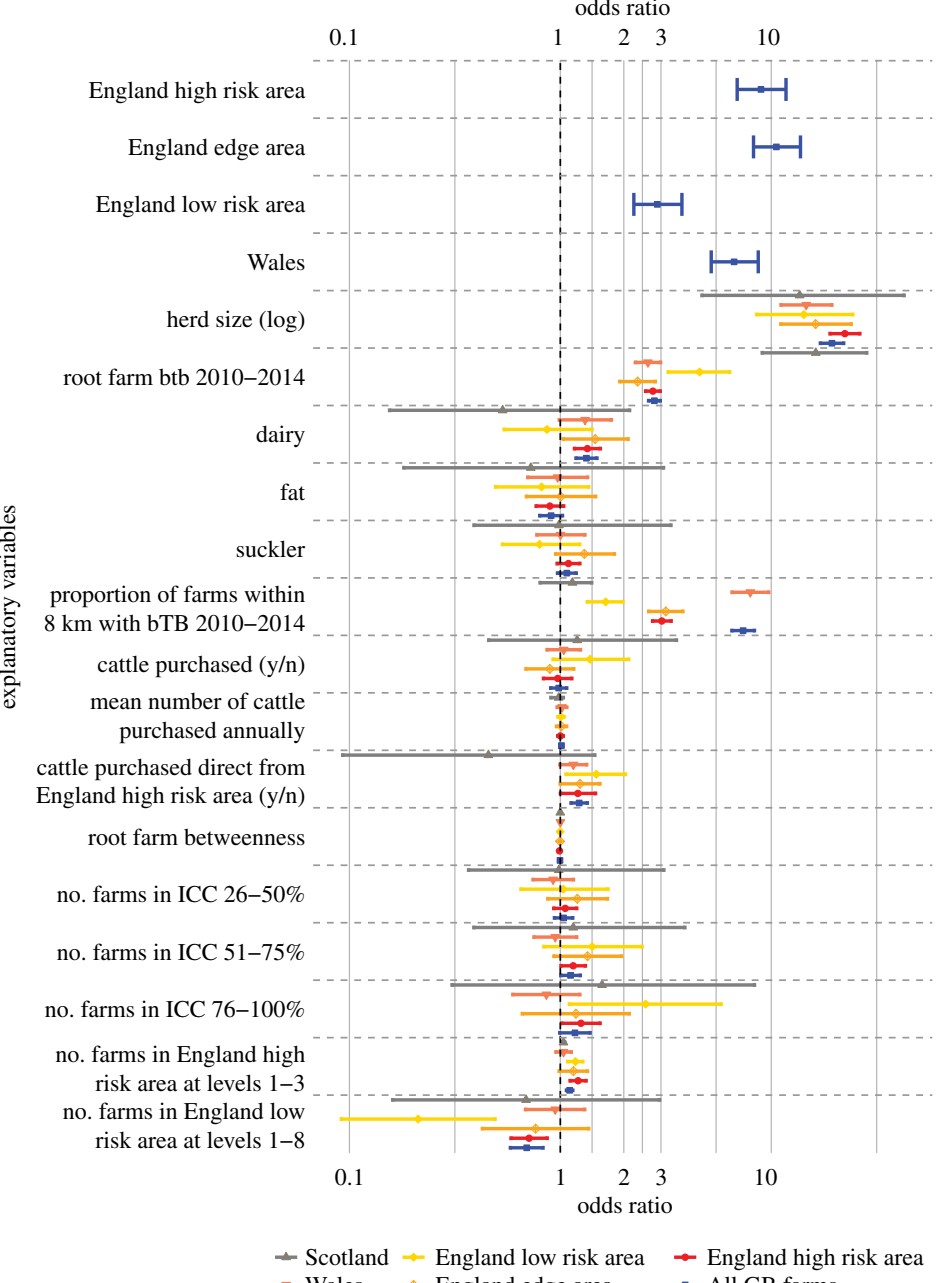

**Figure 3.** The effect of root farm characteristics on the risk of a bTB incident on the root farm in 2015–2016 in Great Britain and in the disease risk regions. Odds ratios of continuous variables are standardized as the odds associated with the difference between the 10th and 90th percentiles of the raw data and are shown with 95% confidence intervals (whiskers). Variables include the number of farms from the England High-Risk Area at levels 1–3 and the number of farms from the England Low-Risk Area at levels 1–8 in the ingoing contact chain of the root farm (ICC), connectivity of the farm within the trading network (betweenness), the number of farms in the ICC and previously established bTB risk factors.

Area decreased the odds of a bTB incident more for root farms in the England Low-Risk Area than for farms in the full Great Britain dataset. In terms of previously established risk factors, the odds of having a repeat bTB incident on root farms in the England Low-Risk Area and in Scotland following an incident in the preceding 5-year period were over fifteen times greater than those for farms which experienced no bTB incident in that period. Similarly, the odds of bTB incidents on farms in the England Low-Risk Area were almost five times greater if they had experienced a previous incident. For root farms in the England High-Risk Area, England Edge Area and in Wales, a repeat incident had more than twice the odds of occurring than a first-time incident. Larger herds had at least thirteen times the odds of experiencing a bTB incident than smaller farms in all regions. Compared to other herd types, dairy

farms still had increased odds of experiencing a bTB incident in 2015–2016 in the England High-Risk Area and England Edge Area. Local effects were strongest in Wales where the proportion of farms within 8 km with previous bTB incidents was associated with almost eight times higher odds (electronic supplementary material, table S2). The effect was lower for farms in the England Low-Risk Area and Scotland, where very few farms had neighbouring farms with recent bTB incidents.

## 4. Discussion

After accounting for known risk factors, we have revealed additional influences on the risks of a new bTB incident on cattle farms arising from the characteristics and bTB history of indirect trading partners in the extensive trading networks of British cattle farms. The risk of a bTB incident increased where there were more farms from the England High-Risk Area at levels 1–3 in the ICC, while having more farms located in the England Low-Risk Area at any level in the ICC was associated with a decreased risk.

Our study covers the period 2010–2016, 4 to 10 years after the introduction of comprehensive pre-movement testing, and selective post-movement testing, and indicates that there remains an increased risk of bTB incidents associated with direct *and* indirect trading with farms in the England High-Risk Area (including trades within the England High-Risk Area). While we found a modest effect (OR = 1.23) of having made a direct purchase from the England High-Risk Area, our results suggest that there is scope for further reductions in the risks of acquiring bTB through trading links. For herds with non-negligible levels of infection, the SICCT test has reasonable sensitivity when considered as a herd-level test [32]; however, when used for pre- and post-movement testing on small batches of cattle (the median number of animals per trade between 2002 and 2015 was 2 with an interquartile range of 1–4 [4]), its sensitivity is likely to be much lower [33], resulting in frequent false negatives and the undesirable movement of truly infected animals. Although farms tend to trade directly most often within their regions [34] and pre- and post-movement testing has been shown to deter movements between regions [17], we show that extensive inter-regional connections are quickly formed in the contact chains of many farms. We observed that within a few movements, large areas of Great Britain might be traversed by cattle movements and that farms can be connected to many farms outside of their risk area or geographical region. While relatively few of these connections are likely to transmit infection, those that do may be effective in translocating disease into new areas [35]. The use of more sensitive diagnostics such as the gamma interferon assay [36] in movement-associated testing, both within and out of annually tested areas, might serve to deter purchase of animals from the England High-Risk Area, due to higher costs, and also to reduce transmission between farms, as the risk of moving infected animals would be reduced. Risk-based trading protocols, including those that quantify direct trading risk, have been outlined elsewhere [37], and including an estimate of the risk associated with indirect trading, using the locations and numbers of source farms might lead to better understanding of the totality of risk associated with each purchase. Data used in this study are readily available for authorities to calculate indirect trading chains of each farm, perhaps to a lesser extent than practiced here, in order to reduce computational time (e.g. up to three movements away from the root farm) and could be then made available to farmers, for example, using an existing application such as the online ibTB tool [38]. Clearly, bTB status is only one of many factors taken into consideration when purchasing stock, if indeed it is considered at all [39]. Focussing solely on sourcing animals based on particular diseases might also unintentionally result in an increased risk of others [40]. Therefore, in addition to bTB specific government controls, holistic measures that reduce the risk of all diseases, such as reducing trade connections and quarantine measures are likely to contribute effectively to herd biosecurity.

Results of the present study show that having more farms in the ICC was initially associated with increased risk, as observed in a similar study of cattle contact chains in France [8]. However, unlike the French study, the number of source farms was not associated with a large increase in the risk of experiencing a bTB incident, suggesting that trade in cattle might play a different role in predicting bTB incidents in the two countries. Bovine tuberculosis prevalence is much lower in France than in England and Wales and there is a comparatively low force of infection from the local environment. This likely results in overall trading strategies being more important, relative to the specific identity of trading partners in the movement network. Our analysis in Britain showed that in terms of bTB incident risk, the location of source farms was more important than the total number of source farms. We predicted that trading with farms that had a history of bTB incidents would be a more accurate predictor of risk than trading with farms from the England High-Risk Area, as reported by Green *et al.* [12]. However, we also found that trading with farms in the England High-Risk Area was more

informative in predicting a subsequent bTB incident on the root farm. This may suggest that past bTB incidents do not fully describe bTB risk and that undisclosed infections in herds and/or environmental sources of infection, such as wildlife, present a risk that is captured by being in the England High-Risk Area but not detected through routine testing [41].

Spatial proximity to farms with recent bTB incidents increased the risk of a bTB incident on the root farm in the full Great Britain model and regional models except for Scotland. We expected that farms closest (i.e. 0–2 km) to the root farm would have the greatest effect on this risk as this would encompass potential contacts of cattle on shared boundaries and the risks from infected local wildlife [42]. However, the variable which best explained this risk was the proportion of farms with a bTB incident within a radius of 8 km. This could potentially be related to the additional effects of unrecorded local cattle movements [12], the spreading of infected slurry [23] and movement of cattle among fragmented land parcels [42]. That this proxy measure still has a large effect size after accounting for other major factors suggests that some or all of these factors may be important drivers of local risk.

Notwithstanding the low number of repeat bTB incidents in Scotland ($n = 21$) and the England Low-Risk Area ($n = 60$) and broad confidence intervals of this variable, the effect of previous incidents in the preceding 5 years (2010–2014) on the risk of a later incident in Scotland or in the England Low-Risk Area was considerably greater than the effect of prior history in the higher risk areas of England and Wales. As the environmental force of re-infection in non-endemic areas is likely to be low, this suggests that a small number of herds in Scotland and the England Low-Risk Area may have had OTF status restored without having fully eliminated infection [41], or alternatively, that there is a repeated external source of infection not accounted for in our study. For example, we have not included cattle imports from outside Great Britain and may therefore have underestimated the risk of bTB on farms that import animals from Northern Ireland and the Republic of Ireland [19], where bTB infection is also endemic in cattle.

An advantage of the regression framework employed in the present study is that it is more straightforward to fit these models to complex data than it would be to fit a fully mechanistic infectious disease model [11,41]. However, while we included approximations for risk based on region and local bTB occurrence, we were unable to fully disentangle the relationships between movements, regions and finer scale spatial risk or the complex interactions between trading, behaviours, herd type and size. It may be that the impact of the network dynamics is more evident if explicit trading paths are modelled, for example, through the use of a compartmental network-based epidemic model [11].

The effects of trading patterns in our study were relatively small compared to other known risk factors such as herd size [43,44] and prior bTB history [8,12,44]. Nevertheless, the major risk factors identified in this and previous studies tend to reflect fixed characteristics of farms (location, past disease history) and those that are difficult, costly or slow to change (herd type). By contrast, elements of trading behaviour, and particular purchasing decisions, might be more amenable to modification, through regulation and incentivization with support from industry and policy-makers. Although certain trading behaviours are clearly related to business type (e.g. dairy, fattening, etc.) [4], careful consideration of existing trading behaviours and potential purchases could result in more robust and safer acquisition and sales of livestock. The present study reinforces the importance of direct trading links but highlights the additional risks associated with indirect trading links, revealed by a more thorough examination of the movement network, as necessary factors to consider in strategies to reduce bTB transmission in Great Britain. In the specific context of bTB and trading, raising awareness of trading risks might prompt farmers to seek more information about their potential trading partners and encourage industry and government to facilitate sharing of this kind of data. Given the computational complexities of interrogating such a dense movement network in real time, the next challenge will be in determining how best these sources of risk might be integrated into current management policies and evaluating how farmers might respond to them.

Further methodological information, figures and tables supporting this article have been uploaded as part of the electronic supplementary material.

Data accessibility. Underlying data consist of every movement of cattle between all farms in Great Britain and their disease history. Aside from the size of the dataset, there are substantial issues of confidentiality (locations, trading practices) and commercial sensitivity in these data. They are collated and managed by Defra, via the Animal and Plant Health Agency, who grant access to the data with specific permissions for specific studies. In practice, this means that the data can be used for the stated purpose only, and making the data publicly accessible would not conform to the licence the authors have been granted to use these data. With the agreement of the journal's Editorial Office, the authors will not be able to make the dataset available on this occasion, but encourage readers, referees and editors to contact the Animal and Plant Health Agency data manager for data access requests. At the time of submission, the data

manager is Andy Mitchell (andrew.mitchell@apha.gov.uk). Code used to run our analysis is available at Dryad Digital Repository: https://dx.doi.org/10.5061/dryad.k0p2ngf56 [45].

Authors' contributions. All authors conceived the study, H.R.F. carried out data analysis; T.J.M. and M.J.S. assisted with data manipulation and analysis; H.R.F., T.J.M. and R.A.M. drafted the manuscript. All authors reviewed the manuscript and gave final approval for publication.

Competing interests. We declare we have no competing interests.

Funding. HRF was funded by an Industrial CASE studentship from the BBSRC grant BB/M015874/1, in partnership with the APHA.

Acknowledgements. The authors gratefully acknowledge Stefan Widgren, Andy Mitchell and the APHA data and epidemiology teams for providing data and assisting with data interpretation.

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
