## [Reviewer comments · Royal Society Open Science]

Review History

RSOS-191806.R0 (Original submission)

Review form: Reviewer 1

Is the manuscript scientifically sound in its present form?

Yes

Are the interpretations and conclusions justified by the results?

No

Is the language acceptable?

Yes

Do you have any ethical concerns with this paper?

No

Have you any concerns about statistical analyses in this paper?

Yes

Recommendation?

Accept with minor revision (please list in comments)

Comments to the Author(s)

General

This paper analyses and assesses the effects of cattle movements in the UK in relation to the spread of bovine Tuberculosis (TB). The paper involves significant computational work, and is clearly related to a current / ongoing policy debate, but I am ambivalent about its value, findings and particularly its recommendations. This is explained below:

Methods

Why does the analysis combine OTFW and OTFS? Why does the analysis take into account the kind or duration of the breakdown? Do the results vary if e.g. short duration breakdowns (mostly OTFS) or 2xIR breakdowns are excluded? I cannot see any reflection on this in the analysis section other than overall summary statistics of the 2 kinds of breakdown. Similarly, it would be incredibly useful for the analysis to exclude 'dead-end' farms - ie those whose stock only went to slaughter. Indeed, there is no mention in the analysis of whether movements to Approved Finishing Units were excluded from the analysis.

Infection is hypothetical in these models, but could the analysis not actually trace reactors/IRs that have moved and subsequently go down with TB? This would represent a more powerful analysis.

Can the authors comment on the limitations of using CPH to represent the actual location of the farm (ln 123). It is well known that CPH's are frequently geographically inaccurate. Although there have been attempts to tidy this up recently, the data used in the paper precedes those attempts. Could the authors comment on how this may/may not affect the results, and/or attempts they have made to limit these errors?

Value/Findings

The final paragraph concludes by suggesting that trading patterns are not that important relative to other risk factors (ln 295). This seems to undo the rather interesting analysis in the rest of the paper, and also raises questions about the value of managing/regulating trading patterns (see below for further discussion). The authors claim that these risk factors (trading) are easier to change in relation to other risk factors. Whether this is true or not (and the authors offer no evidence to suggest it is one way or the other), trading patterns are also indelibly linked to those other risk factors and business models. What would have been more interesting perhaps is for the authors to comment on the methodological difficulty of distinguishing between these different risk factors when there is so much overlap between them. Similarly, some comments on what APHA could do in terms of their data management procedures to help these kinds of analyses could potentially have more impact and be more useful than suggesting that farmers shouldn't buy cattle from the high risk area.

Recommendations

I find the conclusions and recommendations to the paper to not reflect the data studied, nor reflecting the wider literature, particularly from the social sciences.

In some senses, the conclusions seem to contradict the findings of the paper. On the one hand, the authors refer to their study operating in an era of pre/post-movement testing, differentiating it from previous studies. But they also suggest that additional testing requirements are one way which could be used to control the spread of disease. However, most movements are local and within the High-risk areas so it is unclear how additional testing contributes. Are the authors only concerned with movements to the Low Risk Area, or do their recommendations apply to all movements? In fact, Gates' other work in New Zealand also suggests that a significant number of farmers do not take these forms of 'disease rationality' into account when purchasing stock. Other social scientific research in high risk areas suggests that farmers do not take TB-related

factors into account when purchasing cattle. Similarly, recent research from Defra also suggests that TB related factors are not as important as other things such as appearance and temperament. In short: additional testing may make a difference in some circumstances but not all, and other factors are more likely to influence cattle purchasing than TB. Some nuance in the authors' recommendations would be valuable. These conclusions also depend on what the authors mean by 'deter' (ln 247) – whether they are referring to farmers' behaviour arising from statutory regulations, or voluntary risk-based trading measures which they also refer to. Given the data the authors have, it would be useful to show whether changes and/or differences in testing frequencies make any differences in movement patterns and quantify the difference between their study and Salvador et al's paper (currently the authors say they find a smaller effect, but what does this mean?).

The authors also suggest that farmers are less able to change their business models than their purchasing decisions. As suggested above, some evidence to support this argument. As noted above, that there is often significant overlap between the two (in terms of the need to purchase rather than who to buy from), and farmers may be restricted in what they buy either through opportunity, social and economic reasons, it is probably better to steer away from simple recommendations/characterisations of farmer behaviour.

Decision letter (RSOS-191806.R0)

24-Feb-2020

Dear Dr McDonald

On behalf of the Editors, I am pleased to inform you that your Manuscript RSOS-191806 entitled "Effects of trading networks on the risk of bovine tuberculosis incidents on cattle farms in Great Britain" has been accepted for publication in Royal Society Open Science subject to minor revision in accordance with the referee suggestions. Please find the referees' comments at the end of this email.

The reviewers and handling editors have recommended publication, but also suggest some minor revisions to your manuscript. Therefore, I invite you to respond to the comments and revise your manuscript.

- Ethics statement

- Data accessibility

If you wish to submit your supporting data or code to Dryad (<http://datadryad.org/>), or modify your current submission to dryad, please use the following link:
<http://datadryad.org/submit?journalID=RSOS&manu=RSOS-191806>

- **Competing interests**

- **Authors' contributions**

- **Acknowledgements**

- **Funding statement**

Because the schedule for publication is very tight, it is a condition of publication that you submit the revised version of your manuscript before 04-Mar-2020. Please note that the revision deadline will expire at 00.00am on this date. If you do not think you will be able to meet this date please let me know immediately.

If your manuscript is newly submitted and subsequently accepted for publication, you will be asked to pay the article processing charge, unless you request a waiver and this is approved by Royal Society Publishing. You can find out more about the charges at <https://royalsocietypublishing.org/rsos/charges>. Should you have any queries, please contact openscience@royalsociety.org.

on behalf of Dr Dirk Drasdo (Associate Editor) and Kevin Padian (Subject Editor)
openscience@royalsociety.org

Associate Editor Comments to Author (Dr Dirk Drasdo):

Comments to the Author:

We would like you to revise the manuscript in accordance with the reviewer's suggestions. We apologise for the delay in completing review, but an unusually large number of reviewers were required to secure this report. Please ensure you carefully respond to the reviewer's feedback in your revision.

Reviewer comments to Author:

Reviewer: 1

Comments to the Author(s)

General

This paper analyses and assesses the effects of cattle movements in the UK in relation to the spread of bovine Tuberculosis (TB). The paper involves significant computational work, and is clearly related to a current / ongoing policy debate, but I am ambivalent about its value, findings and particularly its recommendations. This is explained below:

Methods

Why does the analysis combine OTFW and OTFS? Why does the analysis take into account the kind or duration of the breakdown? Do the results vary if e.g. short duration breakdowns (mostly OTFS) or 2xIR breakdowns are excluded? I cannot see any reflection on this in the analysis section other than overall summary statistics of the 2 kinds of breakdown. Similarly, it would be incredibly useful for the analysis to exclude 'dead-end' farms – ie those whose stock only went to slaughter. Indeed, there is no mention in the analysis of whether movements to Approved Finishing Units were excluded from the analysis.

Infection is hypothetical in these models, but could the analysis not actually trace reactors/IRs that have moved and subsequently go down with TB? This would represent a more powerful analysis.

Can the authors comment on the limitations of using CPH to represent the actual location of the farm (ln 123). It is well known that CPH's are frequently geographically inaccurate. Although there have been attempts to tidy this up recently, the data used in the paper precedes those attempts. Could the authors comment on how this may/may not affect the results, and/or attempts they have made to limit these errors?

Value/Findings

The final paragraph concludes by suggesting that trading patterns are not that important relative to other risk factors (ln 295). This seems to undo the rather interesting analysis in the rest of the paper, and also raises questions about the value of managing/regulating trading patterns (see below for further discussion). The authors claim that these risk factors (trading) are easier to change in relation to other risk factors. Whether this is true or not (and the authors offer no evidence to suggest it is one way or the other), trading patterns are also indelibly linked to those other risk factors and business models. What would have been more interesting perhaps is for the authors to comment on the methodological difficulty of distinguishing between these different risk factors when there is so much overlap between them. Similarly, some comments on what APHA could do in terms of their data management procedures to help these kinds of analyses could potentially have more impact and be more useful than suggesting that farmers shouldn't buy cattle from the high risk area.

Recommendations

I find the conclusions and recommendations to the paper to not reflect the data studied, nor reflecting the wider literature, particularly from the social sciences.

In some senses, the conclusions seem to contradict the findings of the paper. On the one hand, the authors refer to their study operating in an era of pre/post-movement testing, differentiating it from previous studies. But they also suggest that additional testing requirements are one way which could be used to control the spread of disease. However, most movements are local and within the High-risk areas so it is unclear how additional testing contributes. Are the authors only concerned with movements to the Low Risk Area, or do their recommendations apply to all movements? In fact, Gates' other work in New Zealand also suggests that a significant number of farmers do not take these forms of 'disease rationality' into account when purchasing stock. Other social scientific research in high risk areas suggests that farmers do not take TB-related factors into account when purchasing cattle. Similarly, recent research from Defra also suggests that TB related factors are not as important as other things such as appearance and temperament. In short: additional testing may make a difference in some circumstances but not all, and other factors are more likely to influence cattle purchasing than TB. Some nuance in the authors' recommendations would be valuable. These conclusions also depend on what the authors mean by 'deter' (ln 247) - whether they are referring to farmers' behaviour arising from statutory regulations, or voluntary risk-based trading measures which they also refer to. Given the data the authors have, it would be useful to show whether changes and/or differences in testing frequencies make any differences in movement patterns and quantify the difference between their study and Salvador et al's paper (currently the authors say they find a smaller effect, but what does this mean?).

The authors also suggest that farmers are less able to change their business models than their purchasing decisions. As suggested above, some evidence to support this argument. As noted above, that there is often significant overlap between the two (in terms of the need to purchase rather than who to buy from), and farmers may be restricted in what they buy either through opportunity, social and economic reasons, it is probably better to steer away from simple recommendations/characterisations of farmer behaviour.

Author's Response to Decision Letter for (RSOS-191806.R0)

See Appendix A.

Decision letter (RSOS-191806.R1)

23-Mar-2020

Dear Dr McDonald,

It is a pleasure to accept your manuscript entitled "Effects of trading networks on the risk of bovine tuberculosis incidents on cattle farms in Great Britain" in its current form for publication in Royal Society Open Science.

You can expect to receive a proof of your article in the near future. Please contact the editorial office (openscience_proofs@royalsociety.org) and the production office (openscience@royalsociety.org) to let us know if you are likely to be away from e-mail contact -- if

you are going to be away, please nominate a co-author (if available) to manage the proofing process, and ensure they are copied into your email to the journal.

on behalf of Dr Dirk Drasdo (Associate Editor) and Kevin Padian (Subject Editor)
openscience@royalsociety.org

Appendix A

Fielding et al. Effects of trading networks on the risk of bovine tuberculosis incidents on cattle farms in Great Britain

Reviewer comment	Response
Reviewer: 1 Comments to the Author(s) General This paper analyses and assesses the effects of cattle movements in the UK in relation to the spread of bovine Tuberculosis (TB). The paper involves significant computational work, and is clearly related to a current / ongoing policy debate, but I am ambivalent about its value, findings and particularly its recommendations. This is explained below: Methods Why does the analysis combine OTFW and OTFS? Why does the analysis take into account the kind or duration of the breakdown? Do the results vary if e.g. short duration breakdowns (mostly OTFS) or 2xIR breakdowns are excluded? I cannot see any reflection on this in the analysis section other than overall summary statistics of the 2 kinds of breakdown.	Explanation added (lines 101-105). 'Both types of incident were included in the analysis in order to maximise the power of tests of the association between infection in the contact chain and subsequent incidents. Two additional analyses were performed with each type of incident as the response variable (OTF-S and OTF-W), but final outcomes did not change substantially therefore the simpler, combined analysis is presented here.'
Similarly, it would be incredibly useful for the analysis to exclude 'dead-end' farms – ie those whose stock only went to slaughter. Indeed, there is no mention in the analysis of whether movements to Approved Finishing Units were excluded from the analysis.	Movements to AFUs were included as normal movements in the analysis, however all movements to slaughterhouses were removed as 'dead-end' movements. Herds selling cattle straight to slaughter were included in the analysis, despite them not selling animals to other farms, as they might represent a source of infection into the local environment and are therefore still epidemiologically relevant to assess the risk factors involved in their acquisition of infection. Clearly they do not contribute as source farms.
Infection is hypothetical in these models, but could the analysis not actually trace reactors/IRs that have moved and subsequently go down with TB? This would represent a more powerful analysis.	This is an interesting but wholly different analysis that we have considered and has in part been investigated by one of the authors (McKinley et al., PLOS one, 2018). It is an interesting avenue, which we may address in future papers.
Can the authors comment on the limitations of using CPH to represent the actual location of the farm (In 123). It is well known that CPH's are frequently geographically inaccurate. Although there have been attempts to tidy this up recently, the data used in the paper precedes those attempts. Could the authors comment on how this may/may not affect the results, and/or attempts they have made to limit these errors?	Added lines to explain and justify (lines 132-136). 'There are limitations with this proxy for farm location because, in some cases, the point location of the CPH does not lie within farmland where cattle are mostly located. However, there is no known systematic spatial bias to these locations, sample size is very large and there are no validated alternatives at this scale. Thus, this is the best proxy to use, and it has been used in previous studies [18,26].'
Value/Findings The final paragraph concludes by suggesting that trading patterns are not that important relative to other risk factors (In 295). This seems to undo the rather interesting analysis in the rest of the paper, and also raises questions about the value of managing/regulating trading patterns (see below for further discussion). The authors claim that these risk factors (trading) are easier to change in relation to other risk factors. Whether this is true or not (and the authors offer no evidence to suggest it is one way or the other), trading patterns are also indelibly linked to those other risk factors and business models.	We do not say that they are unimportant, but that the effect sizes are relatively small compared to other known risk factors. We have edited the text at lines 323-325: 'Although certain trading behaviours are clearly related to business type (e.g. dairy, fattening, etc) [4], consolidation and careful consideration of these existing links could result in more robust and safer acquisition and sales of livestock.' See also our response to comments below.
What would have been more interesting perhaps is for the authors to comment on the methodologically	Slightly modified text at lines 312-317: 'However, while we included approximations for

difficulty of distinguishing between these different risk factors when there is so much overlap between them.	risk based on region and local bTB occurrence, we were unable to fully disentangle the relationships between movements, regions and finer-scale spatial risk or the complex interactions between trading, behaviours, herd type and size. It may be that the impact of the network dynamics is more evident if explicit trading paths are modelled, for example through the use of a compartmental network-based epidemic model [11].
Similarly, some comments on what APHA could do in terms of their data management procedures to help these kinds of analyses could potentially have more impact and be more useful than suggesting that farmers shouldn't buy cattle from the high risk area.	Comments added at lines 263-267. 'Data used in this study are readily available for authorities to calculate indirect trading chains of each farm, perhaps to a lesser extent than practiced here, in order to reduce computational time (e.g. up to three movements away from the root farm) and could be then made available to farmers, for example using an existing application such as the online ibTB tool [39].'
Recommendations. I find the conclusions and recommendations to the paper to not reflect the data studied, nor reflecting the wider literature, particularly from the social sciences. In some senses, the conclusions seem to contradict the findings of the paper.	
On the one hand, the authors refer to their study operating in an era of pre/post-movement testing, differentiating it from previous studies. But they also suggest that additional testing requirements are one way which could be used to control the spread of disease. However, most movements are local and within the High-risk areas so it is unclear how additional testing contributes. Are the authors only concerned with movements to the Low Risk Area, or do their recommendations apply to all movements?	We have added text (lines 256-259) to clarify that we are indeed very concerned with movements both within and out of higher risk areas, but that we are particularly discussing the wider use of the more sensitive gamma interferon test. 'The use of more sensitive diagnostics such as the gamma interferon assay [37] in movement-associated testing, both within and out of annually tested areas, might serve to deter purchase of animals from the England High Risk Area, due to higher costs, and also to reduce transmission between farms, as the risk of moving infected animals would be reduced.'
In fact, Gates' other work in New Zealand also suggests that a significant number of farmers do not take these forms of 'disease rationality' into account when purchasing stock. Other social scientific research in high risk areas suggests that farmers do not take TB-related factors into account when purchasing cattle. Similarly, recent research from Defra also suggests that TB related factors are not as important as other things such as appearance and temperament. In short: additional testing may make a difference in some circumstances but not all, and other factors are more likely to influence cattle purchasing than TB. Some nuance in the authors' recommendations would be valuable. These conclusions also depend on what the authors mean by 'deter' (In 247) – whether they are referring to farmers' behaviour arising from statutory regulations, or voluntary risk-based trading measures which they also refer to.	We have added text at lines 267-272. 'Clearly, bTB status is only one of many factors taken into consideration when purchasing stock, if indeed it is considered at all [40]. Focussing solely on sourcing animals based on particular diseases might also unintentionally result in an increased risk of others [41]. Therefore, in addition to bTB specific government controls, holistic measures that reduce the risk of all diseases, such as reducing trade connections and quarantine measures are likely to contribute effectively to herd biosecurity.'
Given the data the authors have, it would be useful to show whether changes and/or differences in testing frequencies make any differences in movement patterns	This is a different analysis that is outside the scope of this paper but would be interesting to explore in a subsequent analysis.
quantify the difference between their study and Salvador et al's paper (currently the authors say they find a smaller effect, but what does this mean?).	On reflection, we can't directly compare the effect sizes from these different analyses as Salvador [19] pertains to the risk of farms in low risk areas buying varying numbers of cattle

	from the high risk area, whereas our analysis separates location of the root farm but then distinguishes whether cattle were bought from the HRA or not from the effect of variation in the number and 'trading distance' of farms in the HRA from which cattle were purchased. Therefore, we have altered our text and removed the qualitative comparison with Salvador (lines 242-243): 'While we found a modest effect (OR = 1.23) of having made a direct purchase from the England High Risk Area'
The authors also suggest that farmers are less able to change their business models than their purchasing decisions. As suggested above, some evidence to support this argument. As noted above, that there is often significant overlap between the two (in terms of the need to purchase rather than who to buy from), and farmers may be restricted in what they buy either through opportunity, social and economic reasons, it is probably better to steer away from simple recommendations/characterisations of farmer behaviour.	We have altered this text a little to clarify (lines 318-325): 'Nevertheless, the major risk factors identified in this and previous studies tend to reflect fixed characteristics of farms (location, past disease history) and those that are difficult, costly or slow to change (herd type). By contrast, elements of trading behaviour, and particular purchasing decisions, might be more amenable to modification, through regulation and incentivisation with support from industry and policy-makers. Although certain trading behaviours are clearly related to business type (e.g. dairy, fattening, etc.) [4], careful consideration of existing trading behaviours and potential purchases could result in more robust and safer acquisition and sales of livestock.' We have also elaborated on this issue in lines 328-333: 'In the specific context of bTB and trading, raising awareness of trading risks might prompt farmers to seek more information about their potential trading partners, and encourage industry and government to facilitate sharing of this kind of data. Given the computational complexities of interrogating such a dense movement network in real-time, the next challenge will be in determining how best these sources of risk might be integrated into current management policies and evaluating how farmers might respond to them.'